# Prevalence of Anti-SARS-CoV-2 Antibodies in Poznań, Poland, after the First Wave of the COVID-19 Pandemic

**DOI:** 10.3390/vaccines9060541

**Published:** 2021-05-21

**Authors:** Dagny Lorent, Rafal Nowak, Carolina Roxo, Elzbieta Lenartowicz, Aleksandra Makarewicz, Bartosz Zaremba, Szymon Nowak, Lukasz Kuszel, Jerzy Stefaniak, Ryszard Kierzek, Pawel Zmora

**Affiliations:** 1Institute of Bioorganic Chemistry Polish Academy of Sciences, 61-704 Poznan, Poland; dlorent@ibch.poznan.pl (D.L.); rnowak@ibch.poznan.pl (R.N.); croxo@ibch.poznan.pl (C.R.); elenartowicz@ibch.poznan.pl (E.L.); a.u.makarewicz@gmail.com (A.M.); bartosz1419@gmail.com (B.Z.); rkierzek@ibch.poznan.pl (R.K.); 2Department and Clinic of Tropical and Parasitic Diseases, Poznan University of Medical Sciences, 60-355 Poznan, Poland; snowak@ump.edu.pl (S.N.); tropiki@spsk2.pl (J.S.); 3Department of Medical Genetics, Poznan University of Medical Sciences, 60-355 Poznan, Poland; kuszel@ump.edu.pl

**Keywords:** SARS-CoV-2, seroprevalence, antibodies, COVID-19, pandemic, Poland

## Abstract

In comparison to other European countries, during the first months of the COVID-19 pandemic, Poland reported a relatively low number of confirmed cases of severe acute respiratory syndrome coronavirus 2 (SARS-CoV-2) infections. To estimate the scale of the pandemic in Poland, a serosurvey of antibodies against SARS-CoV-2 was performed after the first wave of COVID-19 in Europe (March–May 2020). Within this study, we collected samples from 28 July to 24 September 2020 and, based on the ELISA results, we found that 1.67% (25/1500, 95% CI 1.13–2.45) of the Poznan (Poland) metropolitan area’s population had antibodies against SARS-CoV-2 after the first wave of COVID-19. However, the presence of anti-SARS-CoV-2 IgG antibodies was confirmed with immunoblotting in 56% (14/25) samples, which finally resulted in a decrease in seroprevalence, i.e., 0.93% (14/1500, 95% CI 0.56–1.56). The positive anti-SARS-CoV-2 IgG results were associated with age, occupation involving constant contact with people, travelling abroad, non-compliance with epidemiological recommendations and direct contact with the novel coronavirus. Our findings confirm the low SARS-CoV-2 incidence in Poland and imply that the population had little herd immunity heading into the second and third wave of the pandemic, and therefore, that herd immunity contributed little to preventing the high numbers of SARS-CoV-2 infections and COVID-19-related deaths in Poland during these subsequent waves.

## 1. Introduction

The emergence and rapid spread across the globe of severe acute respiratory syndrome coronavirus 2 (SARS-CoV-2) has affected almost every aspect of life. The SARS-CoV-2 virus can cause coronavirus disease 2019 (COVID-19), which is manifested by non-specific symptoms, such as fever, cough, fatigue and rapid loss of taste and smell. The symptoms can range from mild to severe illness [1,2]. In some cases, the SARS-CoV-2 infection can be asymptomatic [3]. Due to the non-specific symptoms or asymptomatic course of novel coronavirus infection, the diagnosis is based on the detection of viral genetic material by the molecular techniques, i.e., reverse transcription and polymerase chain reaction (RT-PCR) or loop-mediated isothermal amplification (LAMP) [4,5]. Currently, the RT-PCR is a gold standard in the diagnosis of SARS-CoV-2 infection due to its high sensitivity and specificity, internal controls of the reaction as well as the possibility of running many samples at once [6,7].

Based mostly on the RT-PCR results, from 1 January to 31 December 2020 the World Health Organization reported almost 81.5 million confirmed cases of SARS-CoV-2 infections and over 1.8 million deaths related to the virus worldwide [8,9]. The most affected countries with highest numbers of new SARS-CoV-2 infections and COVID-19 related deaths among European Union were Italy, Spain, France, Belgium and the UK [10,11]. At the same time, Poland was hit by “two waves” of the pandemic, i.e., from 10 March 2020 to 20 April 2020 and from 4 October 2020 to 27 December 2020, with 1.3 million confirmed SARS-CoV-2 infections and 28.5 thousand deaths caused by the novel coronavirus [12]. Due to the political decisions to test only symptomatic patients [12], nonspecific symptoms of COVID-19 [1,2], and asymptomatic SARS-CoV-2 infections [3], those numbers may be underestimated.

The aims of the present sero-epidemiological study were to estimate the prevalence of SARS-CoV-2 antibodies in the population of the Poznań metropolitan area (Poland) after the first wave of the COVID-19 pandemic, to find the risk factors associated with COVID-19, and to compare the immunoassays used in detection of anti-SARS-CoV-2 antibodies.

## 2. Materials and Methods

### 2.1. Study Design and Participants

We randomly selected and invited 1500 adult (over 18 years old) study participants from approximately 15,000 volunteers living in the Poznań metropolitan area, Poland, who answered the online epidemiological survey. The web-based survey was opened for 5 days, i.e., 18–23 July 2020, and broadly advertised in local and national newspapers, radio, TV, web portals as well as social media. Demographic data, including age, gender and occupation of each participant, were collected. Additionally, we asked volunteers about their current health status, potential flu-like symptoms in the last nine months and behavior during the COVID-19 pandemic, namely compliance with epidemiological recommendations. To estimate the severity of an individual’s flu-like symptoms, we developed a scoring system based on subjects’ self-assessment answers. If the patient did not show any symptoms, a value of 0 was assigned. Each symptom, such as fever, cough, runny nose, fatigue, muscle and joint pain, sore throat, headache, diarrhea and loss of smell or taste, was rated as 1, and hospitalization due to the flu-like symptoms was rated as 5. Additionally, the study participants were asked to compare flu-like symptoms in the last nine months before the serological test to flu-like symptoms in the previous years. Symptoms of flu-like illness in the last nine months that were milder than those experienced in the past were graded as 0, symptoms of the same severity were graded as 3, and more severe symptoms as 5. Based on the total sum of values of individual responses, study participants were divided into four groups: “asymptomatic” (0 points), “mild” (1–5 points), “moderate” (6–14 points) or “severe” (15–24 points). A similar classification approach was implemented to assess individuals’ compliance with epidemiological recommendations. If the study participant did not follow any recommendations, a value of 0 was assigned. Each of the preventive measures, such as face mask use, disinfection and social distance, remote work, avoiding contact with other people, and avoiding the use of public transport, was rated as 1. Depending on the total sum of values of individual responses, study participants were divided into three groups: “non-compliant” (0 points), compliant “at some point” (1–2 points) and compliant “in general” (3–4 points) with epidemiological recommendations.

### 2.2. Laboratory Analysis

The blood samples were collected from individuals from 28 July 2020 to 24 September 2020 (Appendix A) at the Wielkopolskie Centrum Medycyny Podróży, Poznan, and transferred to the IBCH PAS for analysis of anti-SARS-CoV-2 antibodies presence. In case of ELISA positive results for the presence of anti-SARS-CoV-2 IgA or borderline results for the anti-SARS-CoV-2 IgG, the study participants were asked to come again within 2–3 weeks and a blood collection as well as an ELISA analysis were repeated. For five study participants with anti-SARS-CoV-2 IgG-positive results, we did the follow-up analysis at 10 weeks after the first test.

The presence of IgA and IgG antibodies against SARS-CoV-2 was determined using anti-SARS-CoV-2 IgA ELISA (EuroImmun) or anti-SARS-CoV-2 IgG ELISA (EuroImmun) assays, respectively. The chosen immunoassays recognized specific anti-SARS-CoV-2 antibodies against the spike (S) protein and, according to Beavis and colleagues [13], demonstrate good and excellent specificity for IgA and IgG antibodies, respectively. All anti-SARS-CoV-2 IgG positive-samples were confirmed with quantitative anti-SARS-CoV-2 IgG immunoblot (Polycheck), which uses the S protein and phosphorylated nucleocapsid protein (PNC) as antigens.

### 2.3. Statistical Analysis

The categorical variables were presented as counts and percentages, and the seroprevalence estimates were presented together with 95% CI. The 95% CI of the seroprevalence was calculated using the hybrid Wilson/Brown method. The differences between groups were analyzed with Mann–Whitney or Kruskal–Wallis tests. All statistical analyses were performed with the GraphPad Prism 9 software.

### 2.4. Ethics Approval

The study was approved by the Bioethics Committee at the Poznan University of Medical Sciences, Poznan, Poland (Resolution No. 470/20 from 17 June 2019). In addition, written informed consent was obtained from 1500 study participants before blood collection.

## 3. Results

### 3.1. Characteristics of Study Participants

The study group included 1500 adults without any flu-like symptoms at the sampling time and consisted of 896 (59.7%) female and 604 (40.3%) male subjects at a mean age of 38.7 ± 12.7 years old (Table 1). Within the group of volunteers, 964 (64.2%) did not report any chronic diseases, while 536 (35.8%) study participants were treated due to the chronic diseases, such as hypertension, asthma, Crohn disease, rheumatoid arthritis, diabetes or depression. (Table 2). In the last 9 months before the serological tests, 1240 (82.7%) study participants reported flu-like symptoms, i.e., fever, cough, fatigue, muscle and joint pain, etc. (Table 2), mostly with mild (454/1240, 36.6%) and moderate (720/1240, 58.1%) severity. In addition, 356 (23.7%) of volunteers with flu-like symptoms in the 9 last months before blood collection, or with potential contact with a SARS-CoV-2 infected person, were tested for the presence of SARS-CoV-2, and 7 (1.9%) received a positive result (Table 1). Most of the study participants (96.9%) followed the epidemiological recommendations, i.e., wearing a mask covering the nose and mouth, maintaining social distance and regularly disinfecting hands. Only 40 (2.7%) individuals had known contact with a SARS-CoV-2 infected person, while the majority, i.e., 1085 (72.3%) study participants, had not known about such situations (Table 3).

### 3.2. Anti-SARS-CoV-2 Seroprevalence

In total, 25 of 1500 (1.67%) collected samples were found to be positive for anti-SARS-CoV-2 IgG antibodies by the ELISA (Table 1). Simultaneously, 60 of 1500 (4.0%) collected samples were found to be positive for anti-SARS-CoV-2 IgA antibodies, but, in all of those cases, we did not observe IgA to IgG seroconversion (data not shown). There were no significant differences in seroprevalence between the weeks when the blood was collected (data not shown). We did not observe significant differences in the seroprevalence within gender (Table 1). The differences in the anti-SARS-CoV-2 antibodies presence were found among groups at different ages (Table 1) and with different health status (Table 2). Based on the ELISA results, the highest seropositivity was found among people over 65 years old, and among study participants with chronic diseases of the respiratory system (Table 2).

Antibodies against the novel coronavirus were found in only three (42.9%) and eight (2.3%) study participants with positive and negative PCR-based test results, respectively. We found anti-SARS-CoV-2 antibodies in 14 samples from volunteers previously not tested with PCR or antigen tests (Table 1). Additionally, we found that among the tested study participants, 8/25 (35%) of SARS-CoV-2 infections followed an asymptomatic course (Table 2).

### 3.3. SARS-CoV-2 Infection Risk Factors

One of the highest anti-SARS-CoV-2 IgG seroprevalence rates was found among study participants who travelled abroad in the last 9 months before serological tests (Table 3). Others were individuals who did not follow epidemiological recommendations and persons who had direct contact with SARS-CoV-2 (Table 3). Based on the ELISA results, we did not find significant differences in the presence of anti-SARS-CoV-2 antibodies between individuals working in different roles.

### 3.4. Comparison of ELISA and Immunoblot Methods of Anti-SARS-CoV-2 Antibodies Detection

The presence of anti-SARS-CoV-2 IgG antibodies was confirmed with immunoblotting in 14 of 25 samples (56%) (Table 1). For most analyzed parameters, the trend in the anti-SARS-CoV-2 seropositivity was not changed, i.e., based on the immunoblot, study participants over 65 years old, individuals who did not follow the epidemiological recommendations and people with direct contact with SARS-CoV-2 were characterized with highest seroprevalence (Table 1, Table 2 and Table 3). At the same time, the immunoblot analysis revealed that there are significant differences in the seroprevalence between individuals whose occupation involved constant contact with other people, e.g., physicians, nurses, shop assistants and civil servants (Table 3). The ELISA false positive results were found mostly among study participants between 18 and 33 years old (Table 1), as well as among volunteers with chronic diseases of the autoimmunological and respiratory systems (Table 2).

### 3.5. Anti-SARS-CoV-2 Antibody Levels

With quantitative immunoblots, we were able not only to confirm the ELISA results, but also to analyze the levels of anti-SARS-CoV-2 phosphorylated nucleoprotein (PNC) and anti-SARS-CoV-2 spike (S) antibodies. We found different levels of analyzed antibodies among study participants, as shown in Figure 1a. In addition, the anti-SARS-CoV-2 PNC and anti-SARS-CoV-2 S antibody levels differed within the same samples. Due to the low number of positive individuals, we did not correlate the antibody levels with previously mentioned parameters, i.e., age and gender.

With five study participants, we performed follow-up study and analyzed the level of anti-SARS-CoV-2 antibodies at 10 weeks after the first blood collection. As shown in Figure 1b, we did not observe any significant differences, but in 4 of 5 samples we reported slight decrease in the antibody level. For one sample, we found an increase in anti-SARS-CoV-2 S antibody levels (Figure 1b).

## 4. Discussion

For many months, the reported novel coronavirus infection cases and COVID-19-related mortality in Poland were among the lowest in Europe [8,9,11,12]. For example, on 1 October 2020, there were 2469.99 confirmed SARS-CoV-2 cases and 67.20 COVID-19-related deaths per one million citizens, but, at the same time, in Spain and Germany there were 16,652.99 and 3527.39 confirmed infections and 683.84 and 113.49 deaths per one million citizens, respectively [8,9]. Those differences may be explained by the early implementation of public health measures in Poland, such as the closing of primary schools and the so-called deep lockdown in March and April 2020, just after the first confirmed SARS-CoV-2 infections. In addition, the discrepancies in the novel coronavirus cases between Poland and other comparable European Union members may result from the number of performed diagnostic tests. For example, at the beginning of October 2020, an average of 22,125 tests were performed per day in Poland. In comparison, there were 679,134 and 1,123,823 tests performed daily in Spain and Germany, respectively [14]. Finally, the differences in pandemic scale between Poland and other countries may be due to the political decisions to test only symptomatic patients [12] and the lack of free tests for the presence of novel coronavirus for the general population. This excludes individuals with asymptomatic SARS-CoV-2 infection from official statistics and does not prevent emergence of new epidemic foci, since the asymptomatic SARS-CoV-2 infected persons still can infect others [3,15,16]. Furthermore, according to our results, as well as data presented by others, the asymptomatic rate can range from 20% to up to 80% [17,18]. All of the above-mentioned reasons may cause the official numbers of SARS-CoV-2 infections to be underestimated.

Estimation of the scale of the COVID-19 pandemic, as well as objective comparison of different populations, can be achieved through sero-epidemiological studies. Our data demonstrates low (0.93%) seroprevalence of anti-SARS-CoV-2 antibodies in the general population of the Poznan metropolitan area. Poznan is one of the biggest cities in Poland with almost one million inhabitants and, therefore, may represent the situation in other large Polish metropolitan areas, which, in total, represent approximately 30% of the society. However, it should be noted that the seroprevalence can differ in smaller cities and villages, due to, among other factors, the lower access to the health care system and diagnostic centers, as shown by others [19,20]. Based on our results of anti-SARS-CoV-2 seroprevalence, i.e., 0.93%, and demographic data, i.e., data on 3.5 million citizens as well as the numbers of confirmed SARS-CoV-2 infections presented by the Polish Ministry of Health, we calculated that in September 2020, just before the so-called second wave of the pandemic, approximately four-fold more infections occurred than were reported by the government, i.e., 32,550 SARS-CoV-2 infections based on seroprevalence vs. 7985 COVID-19 cases from official statistics. These discrepancies may result from asymptomatic SARS-CoV-2 infection, as mentioned above, as well as reluctance to undergo diagnostic coronavirus testing, which is potentially linked with a mandatory 10-day period of quarantine. The hesitation in testing might result from the lack of trust towards healthcare workers among the general population in Poland. Based on YouGov data, in the context of the ongoing pandemic, the Polish population exhibits higher levels of trust towards family and friends, in contrast to other European nations, who ranked medical experts as the most trusted [21].

In comparison to Poznan, the anti-SARS-CoV-2 seroprevalence at similar time points, in other metropolitan cities, was higher, i.e., in Madrid, Spain, it was equal to 13.6% [22], in Geneva, Switzerland, it was 10.8% [23], and in Tehran, Iran, it was 16.3% [20]. This situation ensured that as of September 2020, the Polish population remained largely immunologically naïve to the virus. The low seroprevalence in Poland also highlights the importance of the vaccination against COVID-19. It is estimated that the spread and transmission of SARS-CoV-2 will be stopped with 60–70% of the population being vaccinated [24]. It is impossible and extremely dangerous to reach this level of herd immunity through SARS-CoV-2 infections.

In addition to estimating the percentage of Poznan metropolitan area citizens that underwent novel coronavirus infection, we also found the SARS-CoV infection risk factors in the wider Polish population. Namely, contact with SARS-CoV-2 infected individuals, age over 65 years, non-compliance with epidemiological recommendations, travelling abroad, and having an occupation involving constant contact with people (i.e., physicians and nurses), are linked with higher seroprevalence. Our data confirmed previously published results [19,25].

Moreover, it should be noted that besides its many strengths, such as the relatively large group of study participants and the use of two independent techniques for anti-SARS-CoV-2 antibody detection, our study has some limitations. First of all, we observed overrepresentation of women in the study, i.e., 59.7% vs. 51.5% in the Polish population, probably due to higher willingness to participate in online surveys [26]. However, since we did not detect significant differences in the seroprevalence among gender, this fact should not bias the final results. In addition, individuals travelling abroad were also overrepresented in the study. According to the Polish Tourism Organisation, 54% of the Polish population travelled for the purposes of vacation in 2019, and 18.3% of them travelled abroad [27]. This fact may lead to some bias and overestimation of seroprevalence. On the other hand, there is underrepresentation of study participants over 65, i.e., 5.1% vs. 22% in Polish society. This fact can be explained by the use of an online survey and the problems encountered by older people when using modern technology [28]. In addition, there were many volunteers who wanted to participate, but due to the COVID-19 pandemic, high risk of severe COVID-19 in this age group, and the recommendation to stay at home and isolate from others, they cancelled the meetings and were replaced by the next random person from the list. Finally, our results can be biased due to the sensitivity and specificity of serological tests. Currently, it was published that many serological tests, including the EuroImmun ELISA used in our study, cannot detect the anti-SARS-CoV-2 antibodies at low levels, which is characteristic for so-called non-responders [29]. This might explain why we did not detect the antibodies in 4 of 7 study participants with positive results in the RT-PCR test. The problem with specificity of serological tests was also observed by us in the case of anti-SARS-CoV-2 IgA antibodies. In all cases of individuals with positive results for IgA antibodies, we did not observe seroconversion. The false positive results of the presence of IgA antibodies were mostly correlated with allergies, which was not described by the manufacturer. It should be also highlighted that ELISA is a very good screening method, but the results should be confirmed by immunoblot. As shown in our study for the first time, the ELISA false positive signals can be as high as almost 50% of all positive results.

## 5. Conclusions

To our knowledge, this is the first study which demonstrates the anti-SARS-CoV-2 antibody seroprevalence in the general population in Poland after the first wave of the COVID-19 pandemic. Our findings confirm that the low SARS-CoV-2 incidence in Poland is probably due to the effectiveness of early countermeasures. However, based on the seroprevalence of 0.93%, it should be noted that the official numbers of novel coronavirus infections were underestimated and that approximately four-fold more infections occurred than were reported by the Polish Ministry of Health. The low anti-SARS-CoV-2 seroprevalence implies that the population had little herd immunity heading into the second and third wave of the pandemic, and therefore, that herd immunity contributed little to preventing the high numbers of SARS-CoV-2 infections and COVID-19-related deaths in Poland during these subsequent waves. Finally, taking into account all above-mentioned limitations of our study, the obtained seroprevalence may be underestimated. Therefore, further studies on the SARS-CoV-2 burden in Poland are needed.

## Figures and Tables

**Figure 1 vaccines-09-00541-f001:**
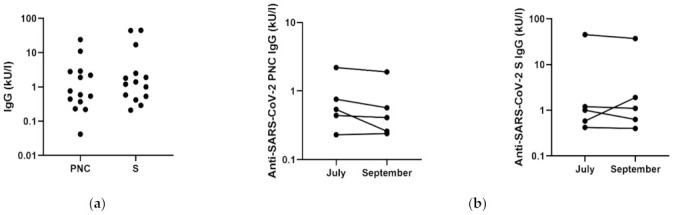
Anti-SARS-CoV-2 antibody levels among volunteers. (**a**) The total levels of anti-SARS-CoV-2 phosphorylated nucleoprotein (PNC) and anti-SARS-CoV-2 spike (S) antibodies among study participants (*N* = 14). (**b**) The changes in the anti-SARS-CoV-2 PNC (**left**) and anti-SARS-CoV-2 S (**right**) antibody levels within 10 weeks in five study participants. Each dot represents a single study participant.

**Table 1 vaccines-09-00541-t001:** Prevalence of anti-SARS-CoV-2 IgG antibodies in Poznań metropolitan area.

Category	Participants	ELISA		Immunoblot	
		Positive Results	Seroprevalence (95% CI)	Positive Results	Seroprevalence (95% CI)
Overall					
	1500	25	1.67% (1.13–2.45)	14	0.93% (0.56–1.56)
Gender					
Female	896	14	1.56% (0.93–2.60)	8	0.89% (0.45–1.75)
Male	604	11	1.82% (1.02–3.23)	6	0.99% (0.46–2.15)
Age					
18–33	623	10	1.61% (0.87–2.92)	5	0.80% (0.34–1.87)
34–49	606	11	1.82% (1.02–3.22)	7	1.16% (0.56–2.37)
50–65	194	2	1.03% (0.18–3.68)	0	0.00% (0.00–1.94)
65+	77	2	2.60% (0.46–8.99)	2	2.60% (0.46–8.99)
Test for the SARS-CoV-2 presence					
Positive	7	3	42.86% (15.82–74.95)	3	42.86% (15.82–74.95)
Negative	349	8	2.29% (1.17–4.46)	6	1.72% (0.79–3.70)
Not tested	1144	14	1.22% (0.73–2.04)	5	0.44% (0.19–1.02)

**Table 2 vaccines-09-00541-t002:** Anti-SARS-CoV-2 seroprevalence related to health status and severity of last flu-like illness.

Category	Participants	ELISA		Immunoblot	
		Positive Results	Seroprevalence (95% CI)	Positive Results	Seroprevalence (95% CI)
Chronic diseases					
None	964	16	1.66% (1.02–2.70)	11	1.14% (0.64–2.03)
CS-CDs	185	1	0.54% (0.03–3.00)	1	0.54% (0.03–3.00)
RS-CDs	78	3	3.85% (1.05–10.71)	0	0.00% (0.00–4.69)
I-CDs	8	0	0.00% (0.00–32.44)	0	0.00% (0.00–32.44)
CKD	1	0	0.00% (0.00–94.87)	0	0.00% (0.00–94.87)
DT-CDs	23	0	0.00% (0.00–14.31)	0	0.00% (0.00–14.31)
A-CDs	170	4	2.35% (0.90–5.76)	1	0.59% (0.03–3.26)
NeoD	5	0	0.00% (0.00–43.45)	0	0.00% (0.00–43.45)
MetD	50	1	2.00% (0.10–10.50)	1	2.00% (0.10–10.50)
MentD	29	0	0.00% (0.00–11.70)	0	0.00% (0.00–11.70)
Severity of flu-like illness in the last 9 months before serological tests					
No symptoms	260	8	3.08% (1.57–5.95)	5	1.92% (0.82–4.42)
Mild	454	5	1.10% (0.42–2.30)	4	0.88% (0.34–2.24)
Moderate	720	12	1.67% (0.96–2.89)	5	0.69% (0.30–1.62)
Severe	66	0	0.00% (0.00–5.50)	0	0.00% (0.00–5.50)

CDs—chronic diseases; CS-CDs—circulatory system CDs, i.e., hypertension; RS-CDs—respiratory system CDs, i.e., asthma; I-CDs—infectious CDs, i.e., HIV/AIDS; CKD—chronic kidney disease; DT-CDs—digestive track CDs, i.e., Crohn disease; A-CDs—autoimmunological chronic diseases, i.e., allergies; NeoD—neoplasmatic diseases, i.e., cancer; MetD—metabolic diseases, i.e., diabetes; MentD—mental disorders, i.e., depression.

**Table 3 vaccines-09-00541-t003:** SARS-CoV-2 infection risk factors.

Category	Participants	ELISA		Immunoblot	
		Positive Results	Seroprevalence (95% CI)	Positive Results	Seroprevalence (95% CI)
Occupation involving constant contact with people (i.e., physicians, nurses, shop assistants, civil servants)					
Yes	749	11	1.47% (0.82–2.61)	9	1.20% (0.63–2.27)
No	751	14	1.86% (1.11–3.11)	5	0.67% (0.29–1.55)
Travelling abroad					
Yes	574	16	2.93% (1.81–4.70)	8	1.46% (0.74–2.86)
No	953	9	0.94% (0.50–1.78)	6	0.63% (0.29–1.37)
Compliance with epidemiological recommendations (i.e., remote work, wearing a mask covering nose and mouth, avoiding contact with other people, avoiding the use of public transport)					
No	46	1	2.17% (0.11–11.33)	1	2.17% (0.11–11.34)
At some point	405	6	1.48% (0.68–3.19)	4	0.99% (0.39–2.51)
Yes, in general	1049	18	1.72% (1.09–2.70)	9	0.86% (0.45–1.62)
Known contact with SARS-CoV-2 infected person					
Yes	40	2	5.00% (0.85–15.79)	2	5.00% (0.89–16.50)
No	375	7	1.87% (0.91–3.80)	5	1.33% (0.57–3.08)
Not known	1085	16	1.47% (0.91–2.38)	7	0.65% (0.31–1.33)

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
