# Peer review of "Prevalence of Anti-SARS-CoV-2 Antibodies in Poznań, Poland, after the First Wave of the COVID-19 Pandemic"

_vaccines, 2021, doi:10.3390/vaccines9060541_

Round 1

Reviewer 1 Report

The authors Lorent et al. have investigated the sero-prevelance of anti-SARS-CoV2 antibodies in general population of Poland. They found the low anti-SARS-CoV-2 2 seroprevalence in Poland which may be the related to the high infections and deaths in Poland. This is paper provides the reasons and importance of vaccinations to reduce the infections. Before the publications, a few comments. 1. The participants in the study have various co-morbidities, how are the co-morbidities affecting the infection with the virus? 2. How are false negatives, false positives test results are contributing to errors in reporting and identifying antibodies? 3. As a future directions, can authors comment on importance of vaccinations even in the people with antibodies. This may be useful piece information for the readers.

Reviewer 2 Report

This perhaps describes the prevalence of SARS-COV-2 in Poland in late 2020. As such, it provides valuable data that can be used by other researchers both in Europe and worldwide.

MAIN ISSUE:   You don't provide a confidence interval for your seropositivity estimate. You have this text in the Discussion: "Currently, it was published that many serological tests, including Euroimmun ELISA used in our study, cannot detect the anti-SARS-CoV-2 antibodies at low levels, characteristic for so-called non-responders (26). This might explain why we did not detect the antibodies in 4 of 7 study participants with positive results of RT-PCR test. The problem with specificity of serological tests was observed also by us in case of anti-SARS-CoV-2 IgA antibodies. In any case of individuals with positive result of IgA antibodies, we did not observed seroconversion. The false positive results of the presence of IgA antibodies were mostly correlated with allergies, which was not described by the manufacturers. And last but not least, it should be highlighted that ELISA is a very good screening method, but the results should be confirmed by immunoblot.

As shown in our study for the first time, the ELISA false positive signals can be as high as almost 50% of all positive results." Could you provide a range of estimates for seropositivity that incorporates any estimates of sampling bias (e.g., from online surveys) and test characteristics as you describe above?  

OTHER ISSUES:  

ABSTRACT don't need the word "true". That is implied by context. Clearly you are not trying to estimate the false scale.   "To estimate the true scale of pandemic in Poland, a serosurvey of antibodies against SARS-CoV-2 was performed after the first wave of COVID-19 in Europe, i.e., between July and September 2020." Break into two sentences so that readers are not confused about (1) when the first wave was, and (2) when you did the seroprevalence sampling.  SO something like: "The first wave of COVID-19 in Europe was between July and September 2020. We collected samples in October and November, 2021."     "had been exposed" do you mean "had antibodies" or do really mean "exposed". The vast majority of exposed people will not develop antibodies. They will have walked by someone in a park and been exposed to a few virions that never even tickled their immune system.    

Does the 56% confirmation rate mean that about half of the initial ELISA results were false positives? Or that half of the followup tests were false negatives?   "Our findings confirm the low SARS-CoV-2 incidence in Poland probably due to the effectiveness of early countermeasures." Your study is not powered to determine why there was a low rate. Can you rule out: (1) biased sampling? (2) test that is not sensitive? (3) other factors that influence reproduction number besides countermeasures? (4) different timing of the wave in Poland compared to other European countries - e.g., later arrival?   "Finally, the low anti-SARS-CoV-2 seroprevalence was one the most important cause of high numbers of SARS-CoV-2 infections and COVID-19 related deaths in Poland during second and third wave of pandemic." I m not so sure that ineffective public health measures were not more important. I would rephrase this to "Finally, the low anti-SARS-CoV-2 seroprevalence implies that the population had little herd immunity heading into the second and third wave of the pandemic, and therefore herd immunity contributed little to prevent the high numbers of SARS-CoV-2 infections and COVID-19 related deaths in Poland during these subsequent waves."  

INTRO "Currently, the RTPCR is a gold standard" just write "Currently, the RTPCR is a standard" A gold standard has 100% sensitivity and accuracy   "At the same time, Poland was hit by “two waves” of the pandemic," give the date range (ideally with exact dates, not just months) for each of these waves. This is important, because it affects the interpretation of the paper. Remember, some readers of this paper may be reading it decades from now, and will not necessarily know the dates.   do you mean "unspecific" or "nonspecific" symptoms? The two words have different meanings.   "test only symptomatic patients (12), unspecific symptoms of COVID-19 (1,2), and asymptomatic SARS-CoV-2 infections (3), those numbers may be underestimated." Or maybe overestimated? Could you explain why each of these would lead to an overestimate or an underestimate?      

METHODS online sampling via questionnaires can result in a highly biased sampling   "answered the online epidemiological survey" how was this surveyed advertised? Who would have known about it? From when to when (dates) was the survey open?   "and transferred to the IBCH PAS for further analysis" what do you mean "further analysis". Had any analysis been done before this?   "an ELISA analysis was repeated" I though this was the first ELISA analysis. You did RT_PCR for the first test, right?   "first test." write either "first RT_PCR test." OR "first ELISA test." depending on what you mean     Maybe cite: Beavis KG, Matushek SM, Abeleda APF, Bethel C, Hunt C, Gillen S, Moran A, Tesic V. Evaluation of the EUROIMMUN Anti-SARS-CoV-2 ELISA Assay for detection of IgA and IgG antibodies. J Clin Virol. 2020 Aug;129:104468. doi: 10.1016/j.jcv.2020.104468. Epub 2020 May 23. PMID: 32485620; PMCID: PMC7255182. and similar papers TO discuss the sensitivity and specificity of the tests you are using, and how the sensitivity and specificity affect your analyses and conclusions.   "In addition, written informed consent was obtained from each study participant." Do you mean everyone (all ~15,000) who took the online survey, or only those who got blood samples taken?   "July and September 2020" can you give us the exact first and last dates of blood collection, otherwise there is a fairly wide range of uncertainty as to exactly when these samples were collected.  

RESULTS "Simultaneously, only 40 (2.7%) individuals had contact with SARSCoV-2 infected person, while majority," I don't think you mean "simultaneously"; perhaps delete this word. Also, you mean "known contact" not "contact".   38% of the 1500 participants traveled abroad recently. Isn't that surprising? What is the expect percentage for Poland? I would expect most members of a population even of a modern European state would not travel abroad very frequently.   "We found significantly different levels of analyzed antibodies among study participants, as shown in Figure 1a." I don't think you mean "significantly". Perhaps just write: "We found different levels of analyzed antibodies among study participants, as shown in Figure 1a."   DISCUSSION Would be interesting to add historical parallels to the Black Death Polish epidemiology to the Discussion. For example, see the discussion here: Why was Poland spared from the Black Death? https://history.stackexchange.com/questions/16699/why-was-poland-spared-from-the-black-death   No way, no plague: was Poland once an island of immunity? https://www.thefirstnews.com/article/no-way-no-plague-was-poland-once-an-island-of-immunity-10943   Did Poland Really Escape the Black Death? https://www.inyourpocket.com/warsaw/did-poland-really-escape-the-black-death_77629f   https://en.wikipedia.org/wiki/Black_Death_migration   Shrouded in Mystery: 6 Myths About the Black Death Plague https://historycollection.com/shrouded-mystery-6-myths-black-death/3/   Historically, plagues might come later to Poland than to Southern and Western Europe. With this in mind, one wonders if Poznan, being centrally located might get hit later than port cities such as Gdansk.       "To our knowledge, this is the first investigation of SARS-CoV-2 seroprevalence in the general population in Poland after so-called first wave of pandemic." I would delete this sentence. I don't think it is necessary. You primarily wrote it to get the attention of the editor/reviewer. The readers don't need to see it once the paper is published.   "reluctance to novel coronavirus diagnostic tests," I would just write "reluctance to coronavirus diagnostic tests," because it isn't clear whether the virus is novel or the tests are novel.   "Based on the YouGov data, the Poles are the only tested population which trust the most family and friends in case of ongoing pandemic (20)." Explain this more. I don't understand what you are trying to say with this sentence.   "This situation ensured that as of September 2020, the Polish population remained largely immunologically naïve to the virus, which explains the dramatic numbers of cases during the second wave of the pandemic" Just write - there are other explanations for the second wave: This situation ensured that as of September 2020, the Polish population remained largely immunologically naïve to the virus."  

FIGURE1A put the number (N) of participants into the legend. i.e., "(N=14)".   FIGURE 1B. "For one sample, we found significant increase in antiSARS-CoV-2 S antibodies level (Figure 1b)." You mean to write "For one sample, we found an increase in antiSARS-CoV-2 S antibodies level (Figure 1b)." You cannot determine significance on one sample.    

FIGURE S1. Update it through the present date (e.g., April 2021). Switch to a 7-day average to get rid of some of the noise. Add in lines for "Europe as a whole" or at least one other representative European Country. The first months in Poland according to reported data were pretty flat, so a comparison with another jurisdiction is needed to show when the waves were in Europe.     GRAMMAR pretty dood, but there are a few issues with tense (verb-subject compatibility, particular with regard to pluralization). Maybe quickly paste into something like Google Docs to check for easily fixable grammatical errors.

Reviewer 3 Report

From 15,000 volunteers in an online survey, 1500 adults were randomly selected and tested for SARS-Cov-2 antibodies. Of the 1.67% participants with positive ELISA results, only 56% were confirmed with immunoblot. Associations with characteristics and risk factors were investigated.  

Please describe (in 2.3 Statistical analysis ) which tests are used for all results.

Please explain why the ELISA test results are presented most prominent (percentage exposed in Abstract, relations with patient characteristics), while 44% of them appear to be false positives.

The known lack of representativeness of the sample according age, and uncertainty according to other characteristics (health status, life style, compliance with recommendations) could be mentioned more prominent in Abstract and Conclusions.

line 17, 19 - Please provide confidence intervals for results in the Abstract.

line 39-45 – Could the reported numbers here be complemented with a date? This would also clearify the difference with the numbers mentioned in the Discussion (with date 1 October 2020).

line 71 – Probably this should be Asymptomatic

Line 110 ‘1500 adults without any flu-like symptoms’ seems to conflict with line 118 ‘… of volunteers with flu-like symptoms’. Please explain.

line 181 – What does ‘significant‘ mean in ‘significantly different levels of analyzed antibodies among study participants’? The same question for line 188 and 189.

line 182 – typo: NPC

line 204-207, 212-214: comparing numbers to numbers in Spain and Germany would be more informative if related to the population size of the countries.

line 235 – Some more numerical information leading to the statement of ‘4-fold more infections’ would be useful.
